# Intestinal Effects of Filtered Alkalinized Water in Lean and Obese Zucker Rats

**DOI:** 10.3390/microorganisms12020316

**Published:** 2024-02-02

**Authors:** Laura Doblado, Ligia Esperanza Díaz, Esther Nova, Ascensión Marcos, María Monsalve

**Affiliations:** 1Instituto de Investigaciones Biomédicas Sols-Morreale (CSIC-UAM), Arturo Duperier 4, 28029 Madrid, Spain; lauradoblado@iib.uam.es; 2Institute of Science, Food Technology and Nutrition (ICTAN), Spanish National Research Council (CSIC), José Antonio Nováis 6, 28040 Madrid, Spain; ldiaz@ictan.csic.es (L.E.D.); enova@ictan.csic.es (E.N.); amarcos@ictan.csic.es (A.M.)

**Keywords:** alkalized filtered water, probiotics, intestine, inflammation, oxidative metabolism, oxidative stress

## Abstract

This study evaluated the intestinal effects of alkalinized filtered water in lean and obese adult Zucker rats. For 3 months, 12-week-old rats consumed either tap water or filtered alkalinized tap water from Madrid city. Weight gain was monitored, changes in metabolism were evaluated by indirect calorimetry, and total antioxidant capacity and levels of inflammatory mediators were measured in plasma. Feces were collected, their microbial composition was analyzed and histological analysis of the small and large intestine was performed, assessing the general state of the mucosa (MUC2), the inflammatory state (F4/80) and the presence of oxidative modifications in protein 4-Hydroxynonenal (4-HNE) by immunofluorescence (IF) and immunohistochemistry (IHC). The results obtained showed that the consumption of alkalinized filtered water improved the composition of the intestinal microbiome and the state of the intestinal mucosa, reducing both local and systemic inflammation and the level of oxidative stress. These changes were accompanied by a better maintenance of the oxidative status in rats. No differences were observed in antioxidant capacity nor in weight gain. The incorporation of probiotics in the diet had a significant impact on the microbiome. These effects were indicative of an improvement in general metabolic, oxidative and inflammatory status.

## 1. Introduction

Gastrointestinal (GI) diseases have an estimated prevalence of 2276.27 million [1]. Among them, functional dyspepsia and acid reflux are particularly common, with prevalence estimations reaching up to 30%, with about 10% reaching Rome IV functional dyspepsia (FD), which incurs considerable associated health impairments [2]. Current pharmacological treatments for FD aim to either neutralize acid or block its production [3]. However, emerging evidence of important side effects associated with long-term treatments have raised concerns and the boosted search for alternative approaches [4]. Among the alternative palliative interventions, drinking alkalinized water has gained considerable attention, especially in Eastern countries. The most commonly used and studied is the electrolyzed alkaline-reduced water (EARW), the consumption of which has been related to an improvement in FD symptoms [5]. The available data also suggest that the consumption of alkalinized water has anti-inflammatory effects, through still-unclear mechanisms that may be of relevance for other pathologies [6]. Of particular interest, in this context, there are other common disorders of the intestinal track such as intestinal dysbiosis.

The industrial revolution fundamentally altered the intestinal microbiome ecosystem, drastically reducing its complexity and increasing its functional instability [7]. A large set of functional studies have demonstrated the key role played by nutritional patterns on the preservation of a healthy microbiome [8] and the marked association of obesity with intestinal dysbiosis [9]. Hence, this evidence has also driven the study of the effect on different water sources, on intestinal dysbiosis and metabolic homeostasis [10]. While it is generally agreed that the pH of drinking water can impact the microbiome, its nature and the effect of EARW consumption are still a matter of controversy. A two-week intervention study with EARW in healthy human volunteers concluded that water pH had no impact on the composition of the gut microbiota or glucose regulation in young male adults [11]. In contrast, in another study, mice were treated for four weeks with EARW, and the authors showed that in the fecal microbiome, the relative abundances of 20 taxa differed significantly from controls. However, the significance of these changes was not stablished [12].

Alternative alkalinization procedures, that do not involve electrolysis but make use of filter-based systems, are also in the market, but scientific studies on their presumed health effects are still at their infancy. We recently evaluated the vascular effects of filter alkalinized water consumption in rodents and found an improved vascular reactivity that was associated with reduced systemic inflammation [13], a result consistent with similar findings on EARW.

Our study aimed to test the intestinal effects of alkalinized filtered water, the possible differential role in the lean healthy vs. obese, using, as a test model, Zucker rats, as well as to assess whether the intake of probiotics, a common treatment for intestinal problems, modifies its intestinal effects. Obese Zucker rats are a widely used model of genetic obesity caused by a mutation (fa) in the gene that encodes the receptor for lectin, an anorexigenic hormone. These rats, which suffer from insulin resistance, glucose intolerance, and metabolic syndrome, are the best existing rat model for studies related to obesity and diabetes [14]. Rats and mice are the most frequently used animal models for biomedical studies, with rats being physiologically, morphologically and genetically closer to humans than mice, which makes them an ideal model for biomedical and pharmacological studies. 

We found that filtered water reduced intestinal inflammation and oxidative stress markers, improved the gut mucosa status, and positively impacted the microbiome profile following three months of treatment.

## 2. Materials and Methods

Rats. Male Zucker lean (lean) and obese (obese) rats were used. The animals were purchased from Charles River and housed at the Instituto de Investigaciones Biomédicas Sols-Morreale (IIBM) animal facility under controlled temperature (20–22 °C) and humidity (50 ± 10%) conditions. The animals were under 12 h light/12 h dark cycles and had ad libitum access to food and water. The animal experimentation protocols were approved by the IIBM Institutional Animal Care and Use Committee, the IIBM Biosafety Committee, the Consejo Superior de Investigaciones Científicas (CSIC) Bioethics Committee and final approval was granted by the Environment Department of the Community of Madrid (Ref. PROEX 117/20). All procedures were in accordance with the Declaration of Helsinki. All animals received humane care in accordance with the criteria described in the “Guide for the Care and Use of Laboratory Animals” prepared by the National Academy of Sciences and published by the National Institutes of Health (No. 86-23 revised in 1985). The used 16-week-old rats were divided into three experimental groups: one group was given tap water (not filtered, not sterilized) available at the institute to drink; the second and third groups were given the same water but previously filtered with the Alkanatur^®^ alkalinizing water filtration system. The water-filtering system was analyzed by an authorized laboratory (Oliver Rodés) and found to conform with AENOR Spanish Norm UNE 149101:2015 for human consumption [15]. The filtering system adds 15 mg/L of Mg^2+^, has been shown to remove Trihalomethanes (THMs) and Cl^−^, to provide water free of microorganisms and to not release Na^+^. The plastic jar has been approved for contact with food and demonstrated to be free of Bisphenol A (BPA), epoxidized soybean oil (ESBO) and phthalate esters [16]. The water was changed daily. The third group of animals was fed a diet supplemented with 3 g/kg of the probiotic Megaflora 9 EVO (Solchem^®^, Barcelona, Spain). Megaflora 9 EVO contains 2 × 10^9^ colony-forming units (cfu)/g of *Bifidobacterium lactis* W51 and W52, *Enterococcus faecium W54*, *Lactobacillus acidophilus* W22, *Lactobacillus paracasei* W20, *Lactobacillus plantarum* W1 and W21, *Lactobacillus salivarius* W24 and *Lactococcus lactis* W19. In humans, the recommended dose is 1–2 g [17]. Each experimental group included 6 animals. Weight gain was monitored every 2 weeks. Fresh feces were collected at *t* = 0, 6 weeks (6w) and three months (3mo) of treatment. The animals were treated for 3mo and then sacrificed by decapitation after intraperitoneal injection with pentobarbital sodium (100 mg/kg). Blood was collected, and plasma and peripheral blood mononuclear cells (PBMC) were separated using a Ficoll-Paque^TM^ PLUS (Cytiva Cytiva Europe GmbH Sucursal en España, Barcelona, Spain) gradient and stored at −80 °C. The intestines were collected, washed in phosphate-buffered saline (PBS) and cut into sections. Tissue sections were snap frozen and stored at −80 °C or fixed in 10% buffered formalin, then embedded in paraffin, and 4 µm sections were cut with a microtome, deparaffinized and hydrated prior to staining.

Indirect calorimetry. The PhenoMaster indirect calorimetry system (TSE Systems GmbH, Bad Homburg vor der Höhe, Germany) was used to analyze metabolism in live, freely moving animals, automatically and continuously, over three light–dark cycles in a dedicated, controlled-environment facility at the IIBM. Rats were first acclimatized to the PhenoMaster room for three days before the experiment began. Each rat was then assigned to an individual cage with a separate gas collection and delivery system, which was controlled by TSE PhenoMaster 5.1 software. The following parameters were examined: volume (V) O_2_, VCO_2_, RER, energy expenditure (H1+), activity (XT + YT), weight of food intake and water volume consumed. Data were registered every hour over 72 h. Animals were exposed to 12 h light and 12 h dark cycles. Data were extracted and analyzed using TSE PhenoMaster 5.1 software. Data presented correspond to the mean values for O_2_ consumed or CO_2_ produced per hour and registered during the day (12 h) or night (12 h) periods of the second 24 h cycle (2nd day), when measurements were more stable.

Microbiome. Feces samples were collected, and bacterial DNA was extracted using the QIAamp^®^ Fast DNA Stool Mini Kit (Quiagen, Hilden, Germany). Bacterial quantification was performed by qPCR using specific primers for each of the tested bacterial groups: *Bacteroides*, *Blautia coccoides-Eubacterium rectale* group, *Clostridium* cluster IV, *Bifidobacterium* spp., *Lactobacillus* spp., Enterobacteriaceae, *Enterococcus* spp., *Faecaibacterium praustnizii* and *Akkermansia muciniphila* directed against the genes that encode 16S rRNA.

Enterobacteriaceae Eco1457 Forward5′-CATTGACGTTACCCGCAGAAGAAGC-3′ Eco1652 Reverse5′-CTCTACGAGACTCAAGCTTGC-3′ *Blautia coccoides*-*Eubacterium rectale* GroupgCcoc Forward5′-AAATGACGGTACCTGACTAA-3′gCcoc Reverse 5′-CTTTGAGTTTCATTCTTGCGAA-3′*Clostridium* cluster IV
sg-Clept Forward 5′-GCACAAGCAGTGGAGT-3′sg-Clept Reverse5′-CTTCCTCCGTTTTGTCAA-3′*Enterococcus* spp.
Enteroc Forward 5′-CCCTTATTGTTAGTTGCCATCATT-3′Enteroc Reverse5′-ACTCGTTGTACTTCCCATTGT-3′*Lactobacillus* spp.Lacto Forward5′-CACCGCTACACATGGAG-3′Lacto Reverse 5′-AGCAGTAGGGAATCTTCCA-3′*Bifidobacterium* spp.Bif Forward5′-TCGCGTC(C/T)GGTGTGAAAG-3′Bif Reverse5′-CCACATCCAGC(A/G)TCCAC-3′*Bacteroides fragilis* GroupBfra Forward 5′-ATAGCCTTTCGAAAGRAAGAT-3′Bfra Reverse 5′-CCAGTATCAACTGCAATTTTA-3′
*Faecalibacterium praustnizii*
Fprau223 Forward 5′-GATGGCCTCGCGTCCGATTAG-3′Fprau420 Reverse 5′-CCGAAGACCTTCTTCCTCC-3′
*Akkermansia muciniphila*
S-St-Muc-1437-a-A-20 Forward 5′- CCTTGCGGTTGGCTTCAGAT-3′S-St-Muc-1129-a-a-20 Reverse 5′-CAGCACGTGAAGGTGGGGAC-3′

Immunohistochemistry (IHC). Tissue fixation and staining were performed using the Vectastain ABC kit and the DAB peroxidase substrate kit (both from Vector Laboratories, Burlingame, CA, USA) following the manufacturer’s instructions. Samples were incubated with anti-F4/80 primary antibody (MCA497; AbD Serotec, Oxford, UK) and then with anti-rat secondary antibody (NA935 GE Healthcare, Boston, MA, USA) linked to alkaline phosphatase (AP), exposed to the AP substrate and, once staining developed, mounted. Images were acquired with a Nikon E90i microscope equipped with a DS-Fi1 camera (Nikon, Tokyo, Japan).

Immunofluorescence (IF). Fixation, staining, and analysis procedures were as previously described [13]. Primary antibodies used were anti-Mucin 2 (sc-7314; Santa Cruz Biotechnology Inc., Dallas, TX, USA) and anti-4-HNE (AB5605, Merk, Darmstadt, Germany). Immunofluorescence was detected by incubation with a fluorescent secondary antibody (FITC; Merck, Darmstadt, Germany). The samples were then counter stained with DAPI (Invitrogen Corp., Carlsbad, CA, USA), mounted, and visualized using a fluorescence microscope, Zeiss LSM 700.

Image analysis. IHC and IF images were analyzed using Fiji-ImageJ 2.0.0-rc-69/1.52p software (NIH, Bethesda, MD, USA) to determine the ratio of positive area to tissue area (IHC) or total integrated fluorescence signal to tissue area (IF).

Gene expression analysis. Intestinal tissue samples were homogenized in the presence of 1 mL of Trizol^TM^ reagent (ThermoFisher Sci., Waltham, MA, USA), and total RNA was isolated following the manufacturer’s instructions. Complementary DNA (cDNA) was synthesized, from total RNA preparations, by reverse transcription of 1 μg of RNA, using Moloney Murine Leukemia Virus (M-MLV) reverse transcriptase (RT) (Promega Biotech Ibérica SL, Alcobendas, Madrid, Spain), in a final volume of 20 μ. The mixture was incubated at 37 °C for 45 min and then cooled for 2 min at 4 °C. The resulting cDNA was used as a template for subsequent quantitative polymerase chain reaction (qPCR). The specific primers used are listed below. Each 10 µL PCR reaction included 1 µL cDNA, 5 µL qPCRBIO SyGreen Mastermix (Cultek SL, Dutchcher Group, San Fernando de Henares, Madrid, Spain) and primers (0.3 µM). Samples were analyzed in triplicate on a Mastercycler^®^ RealPlex2, Eppendorf Iberica SLU (San Sebastian de los Reyes, Madrid, Spain). 36B4 was used as loading control.

*36B4* Forward 5′-GCGACCTGGAGTCCAACTA-3′*36B4* Reverse 5′: ATCTGCTGCATCTGCTTGG-3′*IL-1B* Forward 5′-GCCAACAAGTGGTATTCTCCATGAGC-3′*IL-1B* Reverse 5′-TTGTCACCCCGGATGGAATG-3′*IL-10* Reverse 5′-TTGTCACCCCGGATGGAATG-3′ *IL-10* Forward 5′-GCTCAGCACTGCTATGTTGC-3′

Cytokines. Circulating levels of IL-1β, IL-4, IL-6, TNFα and IL-10 were analyzed by cellular cytometry in plasma samples using Multiplex Cytokine Assays (Pro-cartaPlex Immunoassays (ThermoFisher Scientific, Waltham, MA, USA)) at the Centro Nacional de Biotecnología (CNB, CSIC, Madrid, Spain) Flow Cytometry Unit. The lean rat samples were pooled in pairs (each analyzed sample/data point is the mixture of two plasma samples, from two rats); obese rat plasma samples were not pooled. The values obtained were corrected by the concentration of proteins in the sample determined by the Lowry method via colorimetry.

Antioxidant capacity. Antioxidant capacity was determined in plasma samples using the e-BQC electrochemical analytical system (BioQuoChem, Oviedo, Spain), which measures total (QT), fast (Q1) and slow-acting (Q2) antioxidant capacity.

Statistical analysis. Microsoft Excel 16.16.27 (Microsoft, Redmond, WA, USA) was used for data processing. GraphPad Prism 9 (Dotmatics, Boston, MA, USA) was used for statistical analysis and graph preparation. Data are expressed as mean ± standard deviation (SD) or standard error of the mean (SEM), as indicated in the figure legends. The normality of the data was assessed using the Kolmogorov–Smirnov test. Statistical significance of differences between groups was assessed using two-tailed unpaired or paired *t*-test for comparisons between two groups; one-way and two-way ANOVA was used for the analysis of time-dependent variation and the differential response to treatments of two groups, respectively. Levene’s test was used for equality of variances. The values were considered statistically significant when *p* < 0.05. No values were discarded; values not included could not be obtained due to acquisition problems or technical problems with the samples or animals.

## 3. Results

In order to test the intestinal effects of alkalinized filtered water and the differential impact in obesity, lean and obese Zucker rats were treated for 3mo with tap water or filtered water. The probiotic Megaflora 9 EVO was included in the diet of a group of rats dinking filtered water, adjusting the dose to that recommended for humans. The decision to perform a 3mo treatment (3mo) was based on our previous studies, showing significant changes in rats treated for 3mo. We also collected blood and stool samples at 6 weeks for a better follow-up of the treatment responses.

### 3.1. Weight Gain

The rats’ weight gain was monitored every two weeks, and both absolute and relative weight gain values were determined (Figure 1). The results did not show significant differences associated with the treatments except at four weeks of treatment, when the relative weight gain (%) of the filtered water and probiotic groups was significantly greater than that of the filtered water group, but this difference was not maintained at longer treatment times. Similarly, the areas under the curve (AUC) for the absolute values did not show significant differences between treatment groups. Therefore, we concluded that the treatments did not have a relevant effect on the animals’ weight gain.

### 3.2. Circulating Cytokines

Next, in order to determine the general inflammatory status of the rats, the circulating levels of cytokines (IL-1β, IL-4, IL-6, TNFα and IL-10) were evaluated in blood plasma samples. Significant changes were found in the levels of IL-1β and IL-10, cytokines generally considered as markers of type M1 and M2 macrophages, respectively. After 6w of treatment, higher levels of IL-1β were observed in lean rats that consumed tap water than in rats that consumed filtered water, and lower levels of IL-10 in rats that consumed tap water than in rats that consumed filtered water (Figure 2), suggesting that drinking alkalinized water favors an immunosuppressive and anti-inflammatory profile.

### 3.3. Antioxidant Capacity

Then, we evaluated how the treatments affected the total antioxidant capacity in blood plasma samples. To this end, fast (Q1), slow (Q2) and total (QT) antioxidant capacities were measured using the e-BQC electrochemical reader (BioQuoChem). No significant differences were observed between the treatment groups at either 6w or 3mo (Figure 3), suggesting that the treatments did not globally affect the antioxidant capacity of the rats.

### 3.4. Indirect Calorimetry

In order to analyze the impact of the treatments on systemic metabolism, an indirect calorimetry system was used to determine the rats’ O_2_ consumption and CO_2_ production and how they changed in response to treatments at 6w (1.5mo) and 3mo (Figure 4). The group of tap water rats showed a gradual decrease in metabolic activity, affecting both O_2_ consumption and CO_2_ production. These changes were significant in the night period in normal rats at 3mo and in obese rats both during the night and day periods at 3mo. The decrease in CO_2_ production was also significant in the obese tap water group of rats at 6w. These reductions were not observed or did not reach statistical significance in the groups treated with filtered water, both with and without probiotics. In fact, in obese rats, the magnitude of the difference was significantly smaller in rats treated with filtered water and probiotics than in control rats, in the day and night periods, for both O_2_ consumption and CO_2_ production. These results suggest that treatment with filtered water, particularly in combination with probiotics, maintains a higher metabolic activity than tap water, especially in obese rats.

### 3.5. Intestinal Mucosa

#### 3.5.1. Mucin

In order to evaluate the intestinal impact of the treatments, we studied the intestinal mucosa of the small intestine and colon through the IF labeling of intestinal tissue sections with antibodies directed against mucin (MUC2), an oligomeric protein that is a fundamental component of the intestinal mucus. The reduction in MUC2 levels is generally associated with the loss of the intestinal barrier, increasing the risk of inflammatory processes and intestinal infections [18]. In lean rats, treatment with filtered water significantly increased mucin levels in the colon. Likewise, the global comparison of all the lean rats treated with filtered water (with or without probiotic), with respect to the tap water group, showed a significant increase in mucin levels in the colon. Furthermore, in obese rats, a significant increase in mucin levels was observed in rats treated with filtered water and probiotics compared to the tap water group. The difference was also significant when rats treated with probiotics were included in the analysis (Figure 5). These results suggest that treatment with filtered water, with or without probiotics, results in a better maintenance of the intestinal mucosa than tap water.

#### 3.5.2. Intestinal Inflammation

F4/80

Considering that changes in the mucosa are usually related to the inflammatory state and that the results pointed to an improvement in the inflammatory state at the systemic level, in rats treated with filtered water, we established the level of intestinal inflammation by determining the presence of macrophages by IHC staining with an F4/80 antibody.

Analysis of the F4/80-positive area showed, in the small intestine of normal rats and in the colon of obese rats, a significantly lower presence of macrophages in rats treated with filtered water than in rats that consumed tap water. Rats treated with filtered water and probiotics showed a similar trend, but, in this case, the differences did not reach statistical significance (Figure 6). These data are consistent with the improvement observed in the intestinal mucosa and the reduction in inflammation at the systemic level, observed previously, supporting that rats treated with filtered water had a better intestinal status.

Cytokine’s gene expression

Subsequently, the expression levels of *IL-1B* and *IL-10* in the intestinal tissue were analyzed. The mRNA levels of *IL-1B* in the small intestine of obese rats treated with filtered water and probiotics were lower than those of tap-water-treated rats. However, regarding the other groups, no significant changes were found, although obese rats treated with filtered water showed a trend towards lower levels of *IL-1B* and higher levels of *IL-10* than rats treated with tap water, both in the small and large intestine sections, an observation consistent with the data derived from plasma samples (Figure 7).

Oxidative stress (OS)

To determine if these changes were related to the levels of oxidative stress present in the tissue, the presence of proteins modified with 4-HNE was analyzed by IF, using specific antibodies. Rats treated with filtered water had, overall, significantly less modified proteins in the small intestine than tap water rats. The same trend was observed in the colon, as well as in rats treated with filtered water and probiotics, although differences did not reach statistical significance (Figure 8). Therefore, these data support the idea that treatment with filtered water reduces oxidative stress at the intestinal level, an effect that may be related to an improvement in the inflammatory status and the preservation of the mucosa.

### 3.6. Intestinal Microbiome

Finally, the impact caused by filtered water, by itself or in combination with probiotic, on the intestinal microbial composition was evaluated. To that end, fresh feces were collected prior to the intervention, at 6w and at 3mo of treatment. DNA was isolated, and bacterial groups were analyzed by qPCR. The bacterial groups studied have been previously shown to be altered in subjects with metabolic syndrome [19], obesity [20], diabetes [21], and/or to undergo strong variations in response to diet type. In particular, *Bacteroides fragilis* group, *Blautia coccoides-Eubacterium rectale* group, *Clostridium* cluster IV, *Bifidobacterium* spp., *Lactobacillus* spp., Enterobacteriaceae, *Enterococcus* spp., *Faecaibacterium praustnizii* and *Akkermansia muciniphila* were studied. Although the variety of microbiomes is as great as the subjects that carry them, and the diets that feed them, general trends associated with poor intestinal function, dysbiosis, have been identified, which is characterized by four phenomena that can be concurrent, the loss of beneficial microbiota, the overgrowth of potentially harmful bacteria [22], a general reduction in microbial diversity [7] and important changes in the metabolism of the bacteria themselves, as evidenced by the increased or decreased presence of products of their catabolism [23]. In particular, a significant decrease in the bacteria *Akkermansia muciniphila* and *Faecalibacterium praustnizii* has been observed in obese people when compared to people with normal weight [24,25]. The data obtained in our study for *Akkermansia muciniphila* showed a significant difference between the groups after 3mo of treatment. While in normal rats, there was a significant decrease after 3mo of treatment with respect to basal levels (at *t* = 0), this reduction was of a significantly lesser magnitude (filter + probiotic) or did not occur (filter) in the other groups. The same trends were observed in obese rats, reaching statistical significance for the comparison between rats treated for 3mo with filtered water and rats treated with tap water. Regarding *Faecalibacterium praustnizii*, the magnitude of the changes was much smaller, and only a non-significant trend was observed towards a decrease in obese rats treated with tap water at 6w, which reached statistical significance in the group of rats treated with filtered water and probiotics, but no significant differences were observed between groups. Taken together, these data suggest an improvement in these groups derived from treatment with filtered water that is detectable in both normal and obese rats.

The *Clostridium* cluster IV, to which *Faecalibacterium praustnizii* belongs, is considered beneficial for its ability to produce butyrate, and its levels generally increase in response to diets rich in nutrients of vegetable origin [26]. In the present study, *Clostridium* cluster IV was markedly modified in response to treatments. In the group of rats that were treated with tap water, we observed, over time, a significant decrease in *Clostridium* cluster IV levels that was not observed in other groups. Differences among groups reached statistical significance for obese rats at 6w comparing the tap water group with the group treated with filtered water and for the levels at 6w and 3mo for the comparison of the control group with the group treated with filtered water and probiotics. In addition, a significant difference was also found for the comparison of obese rats treated with filtered water and probiotics with those treated with filtered water at 3mo, with levels being higher in rats treated with filtered water and probiotics. Therefore, in this context, both filtered water and probiotics tended to increase or maintain higher levels of *Clostridium* cluster IV.

*Clostridium* belongs to the Firmicutes phylum, which includes over 200 genera, including *Lactobacillus* and *Bacillus*. Both an increase in Firmicutes and a decrease in species of the Bacteroidetes phylum are commonly considered indicators of microbial dysbiosis associated with metabolic syndrome [8]. Belonging to the Clostridia class, the *Blautia coccoides-Eubacterium rectale* group is particularly related to dysbiosis associated with obesity and metabolic syndrome [27]. In the context of the present study, a significant reduction in these bacteria was observed in tap water rats at 6w of treatment, in normal rats, and at 3mo, in obese rats. Also, the group treated with filtered water and probiotics produced a significant decrease in normal rats, at 3mo, and in obese rats, at six weeks, but no significant differences were found between the groups.

Regarding *Lactobacillus* spp., the currently available studies indicate that different species of the genus behave differently in relation to obesity and metabolic syndrome. Similar to observations regarding *Clostridium* cluster IV, there is a strong modification in response to the type of diet, with levels varying greatly depending on the consumption of diets enriched in sugar or fat [20]. In the context of the present study, we observed that obese rats had higher levels of *Lactobacillus* spp. than the group of tap water rats (7.12+/−0.31 vs. 6.89+/−0.26 Log10 CFU/gr feces), and normal rats treated with filtered water and probiotics presented a significant reduction in the levels at 3mo. This change was not observed in the other groups or in the obese rats, suggesting that this potentially beneficial effect is dependent on the treatment with probiotics and the absence of obesity.

Bacteria located in the distal intestine, such as Bacteroides, Bifidobacterium and Enterococcus, are essential because they synthesize vitamins such as B12 and K [28]. *Enterococcus* spp. has been shown to maintain an inverse correlation with weight gain [29]. In this study, in normal rats, the group treated with filtered water and probiotics increased the levels of *Enterococcus* spp. both at 6w and 3mo, and this increase was significant when making a comparison within the group as well as when comparing the groups with each other, suggesting that this effect could be caused by probiotic consumption. However, in obese rats, both the tap water group and the groups treated with filtered water, with or without probiotics, showed a decrease compared to basal levels that was significant in all cases at 3mo, with no significant differences detectable among the groups, suggesting that the effect of the probiotic is lower in obese rats. Although obesity has been related to a loss of Bacteroides in general and *Bacteroides fragilis* in particular [30], the fecal concentration of *Bacteroides fragilis* has been shown to positively correlate with body mass index in some cases [31]. In this study, it was observed that lean tap water rats and those treated with filtered water and probiotic had a significant decrease in *Bacteroides fragilis* levels at 3mo of treatment. This change was not observed in rats treated with filtered water, but the comparison with the control group did not reach statistical significance. In obese rats, an increase was observed after 3mo in rats treated with filtered water and probiotics, which was not detectable in the other groups. These data could be indicative of an improvement in relation to Bacteroides fragilis, which in normal rats would be mainly associated with filtered water, while in obese rats, it would also require probiotics.

*Bifidobacterium* spp. are also generally found reduced in obese individuals [24]. In the current study, significant changes were only observed in response to the treatments in obese rats, in which, at 3mo, the group treated with filtered water and probiotics showed higher levels when comparing both within the group at *t* = 0 and between groups, which suggests that it is an effect dependent on probiotics and only detectable in obesity.

The Proteobacteria phylum, Gram-negative bacteria, is the one that has been most consistently associated with obesity and metabolic syndrome. Specifically, Enterobacteriaceae have been found increased not only in feces but also in blood and various tissues in this context [32]. In this study, a significant increase in these bacteria was found in normal rats treated with tap water, both after 6w and 3mo of treatment but not in rats treated with filtered water or filtered water and probiotic. Furthermore, the contrast between control rats and rats treated with filtered water was significant after 6w of treatment, and the contrast between control rats and rats treated with filtered water at 3mo. No significant changes were found in obese rats. Therefore, these data seem to indicate that the treatment with filtered water prevents the increase in these bacteria in lean rats.

## 4. Discussion

The high prevalence of FD has boosted the search for palliative interventions. The use of EARW has been proposed to improve FD symptoms [5], suggesting that the consumption of alkalinized water may have more general health-promoting effects [6]. However, the available data have not consistently supported this hypothesis [11]. Our group performed a previous study that tested the effects of the consumption of filtered alkalinized water in mice and rats for 3mo and found evidence suggesting a reduction in oxidative stress and inflammation [13]. The present study, thus, aimed to evaluate the intestinal impact of filtered alkalinized water in lean and obese rats. The results obtained support the hypothesis that filtered alkalinized water, when compared to unfiltered–unsterilized tap water from Madrid city, has a positive effect on both systemic and intestinal inflammation, better preserving the intestinal mucin layer and preventing intestinal dysbiosis. These changes were linked to a better maintenance of oxidative metabolism, suggesting a better metabolic status of the rats consuming filtered water (Figure 4). Of note, the improved metabolic profile is not related to any detectable changes in weight gain (Figure 1). Taking into consideration the detected changes in the microbiome, with increased *Akkermansia muciniphila* and *Clostidium* cluster IV and decreased Enterobacteriaceae in response to treatment with filtered alkalinized water (Figure 9), it is tempting to speculate that these changes are likely the main drivers of the preservation of the intestinal mucin layer, leading, in turn, to reduced intestinal and systemic inflammation, with a general positive impact on whole body metabolism. *Akkermansia muciniphila* is a mucin-degrading bacterium that has also been shown to stimulate mucin production with a net positive impact on the thickness of the mucin layer, the preservation of gut barrier function and intestinal immune homeostasis [33]. *Clostridium* cluster IV is well known for efficiently fermenting the plant polysaccharides composing dietary fiber. Clostridium species have been reported to attenuate inflammation, and its metabolites, including butyrate, secondary bile acids and indolepropionic acid, have been shown to play a probiotic role, primarily through their effects on intestinal epithelial cells, strengthening the intestinal barrier [34]. Enterobacteriaceae are facultative anaerobes that, in normal conditions, play a key role in intestinal oxygen consumption, allowing for the proliferation of anaerobes, necessary for polysaccharide fermentation. Nutrient overload and exposure to inflammatory mediators have been proposed to reduce their oxidative capacity and drive their proliferation, thus promoting obesity-related dysbiosis [35].

Therefore, these changes in intestinal bacteria are consistent with the observed increased levels of mucin in the intestine (Figure 5) and the reduced presence of macrophages (Figure 6), indicating a better status of the intestinal mucosa and a lower level of inflammation also detectable at the systemic level, since a reduction in IL-1β levels together with an increase in IL-10 levels was found in plasma samples (Figure 2). Consistently, both the reduced inflammation and a better oxidative status were associated with reduced intestinal oxidative stress, as evaluated by the presence of 4-HNE-modified proteins in the intestine (Figure 8). The reduced oxidative stress is not likely to be associated with changes in antioxidant activities since the overall antioxidant capacity in plasma samples was not significantly changed (Figure 3).

The observed changes in response to the treatments were generally observed in both lean and obese rats, even if the magnitude/significance varied, suggesting a similar response pattern in both cases.

Although highly tentative at this stage, the derived data suggest that alkalinized filtered water is possibly less irritating to the intestinal mucosa than tap water, and it may also have a similar direct effect on the intestinal bacteria, helping to preserve intestinal homeostasis, hence preventing systemic inflammation and boosting metabolic homeostasis.

The diet supplementation with probiotics had a distinct impact on the microbiome that was clearly independent from the effect of the water, although further testing of a group treated only with the probiotic would be necessary to draw definite conclusions (Figure 9). Nevertheless, since no clear indication of a cooperative or cancelling cross-effect was identified, it possibly suggests that alkalinized filtered water and the tested probiotics can be used safely in combination.

Therefore, these data support the hypothesis that drinking filtered alkalinized water can improve the intestinal inflammatory state and reduce dysbiosis, resulting in a better state of general health, although human studies will still be necessary to confirm the significance of these findings.

## Figures and Tables

**Figure 1 microorganisms-12-00316-f001:**
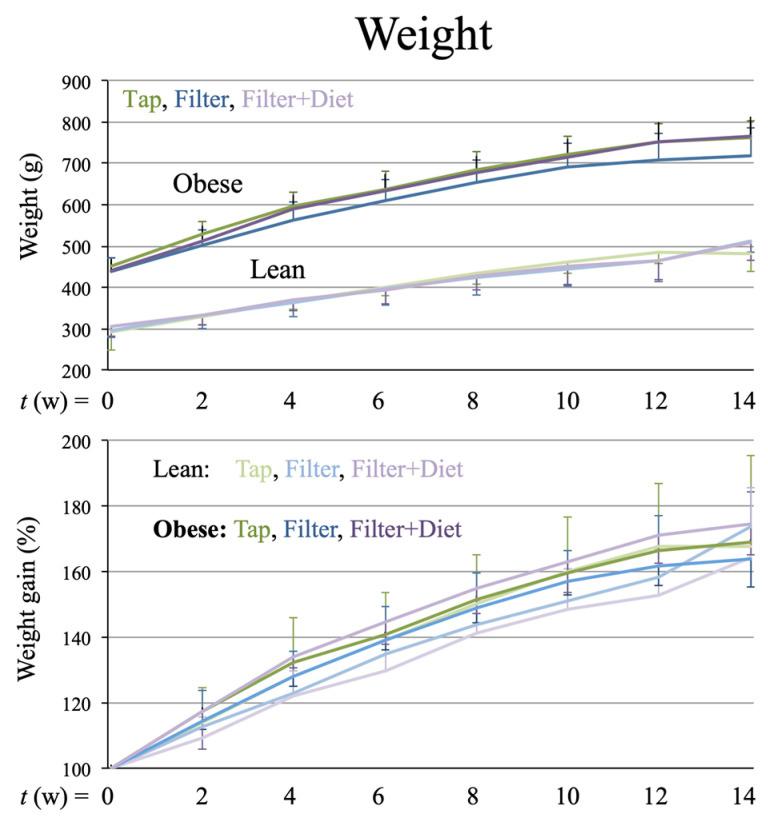
Absolute (**upper panel**) and relative (**lower panel**) weight gain determined every 2 weeks for 3mo. In the bottom panel, weight values at *t* = 0 were considered 100%. The graphs show mean +/− SD.

**Figure 2 microorganisms-12-00316-f002:**
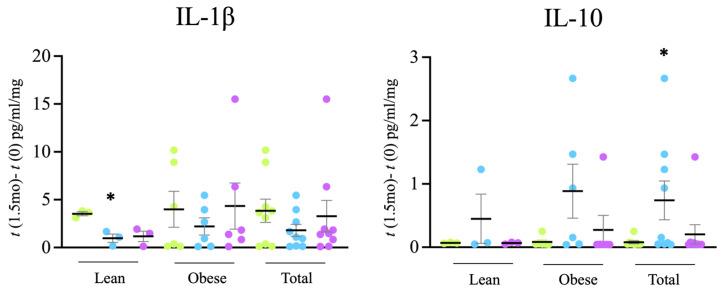
Changes in plasma levels of IL-1β and IL-10 (levels at 6w—levels at *t* = 0) following 6w of treatment. 
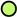
 Tap, 
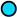
 filter, 
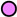
 filter + probiotic. The graph shows mean +/− SEM. * *p* < 0.05 (*t* test).

**Figure 3 microorganisms-12-00316-f003:**
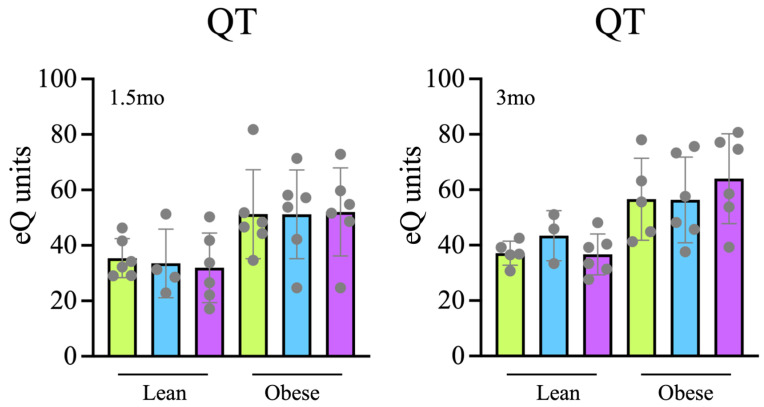
Electrochemical analysis of the total antioxidant capacity (QT) in plasma samples from rats treated for 6w (**left panel**) or 3mo (**right panel**). 
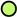
 Tap, 
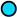
 filter, 
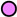
 filter + probiotic. The graph shows the electrochemical units provided by the e-BQC reader and includes mean values +/− SD.

**Figure 4 microorganisms-12-00316-f004:**
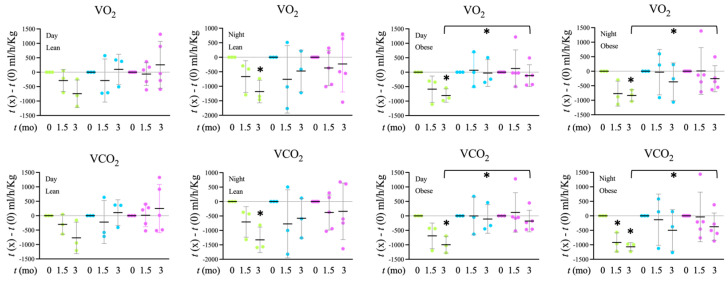
Analysis of O_2_ consumption (**upper panels**) and CO_2_ production volumes (**lower panels**) by indirect calorimetry. The rats were evaluated before treatment, at 6w and 3mo of treatment. 
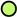
 Tap, 
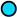
 Filter, 
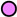
 Filter + Probiotic. The values shown in the graphs are the difference of O_2_ volume (V) consumed (**upper panel**) or CO_2_ produced (**lower panel**) between the indicated time and *t* = 0 for each individual rat analyzed. Determinations were taken every hour for 72 h, the presented data corresponds to the mean of both 12 h day and 12 h night periods of the 2nd day, when collected values were more stable. The graph shows mean values +/− SD. * *p* < 0.05 (*t* test).

**Figure 5 microorganisms-12-00316-f005:**
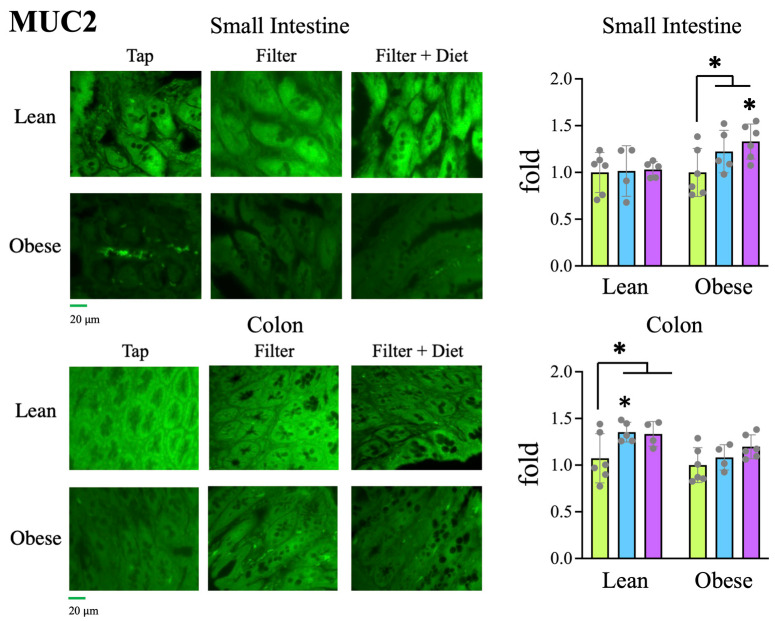
Determination of MUC2 levels by IF analysis of tissue sections from the small intestine and colon. 
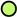
 Tap, 
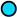
 filter, 
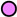
 filter + probiotic. Collected data correspond to the integrated fluorescence signal divided by the tissue area. The graphs show the x fold change relative to the mean of the lean or obese control group (tap), mean +/− SD. * *p* < 0.05 (*t* test). Each data point corresponds to the mean values per rat. Representative images of the groups taken with a 20× objective are included in the **left panels**.

**Figure 6 microorganisms-12-00316-f006:**
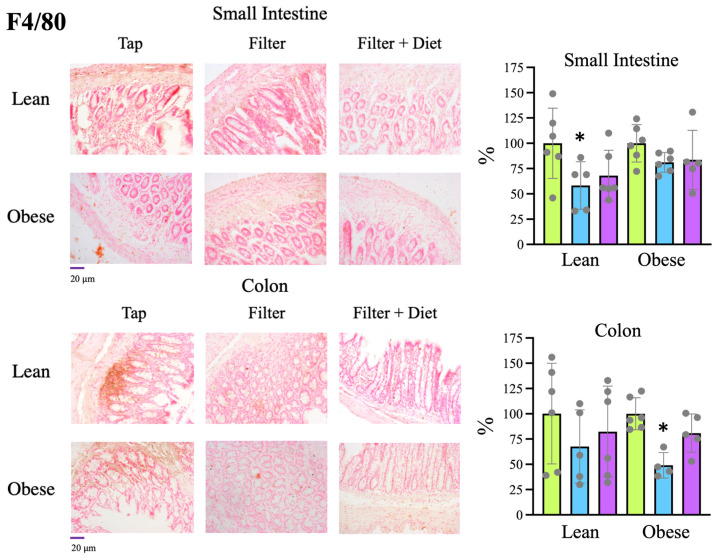
IHC staining of macrophages in tissue sections from the small intestine and colon using antibodies directed against F4/80. 
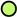
 Tap, 
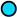
 filter, 
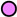
 filter + probiotic. Collected data correspond to the positive area divided by the tissue area. The graphs show the % relative to the mean of the control group (tap), mean +/− SD. * *p* < 0.05 (*t*-test). Each data point corresponds to the mean of the values obtained for each rat. Representative images of the groups taken with a 20× objective are included in the left panels.

**Figure 7 microorganisms-12-00316-f007:**
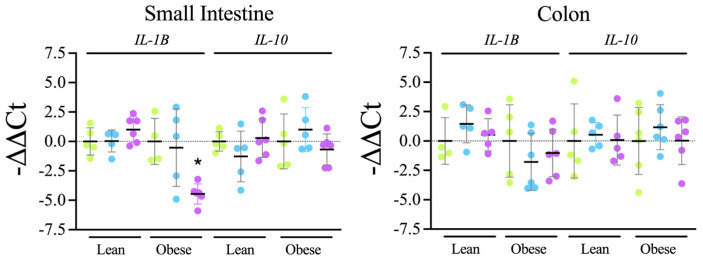
qRT-PCR analysis of the expression levels of *IL-1B* and *IL-10* in small intestine and colon tissue samples. 
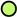
 Tap, 
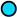
 filter, 
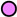
 filter + probiotic. The graphs show the differences in ΔΔCt of the individual values per rat and the mean value of the corresponding control group (tap), mean +/− SD. For clarity’s sake, −ΔΔCt values are presented, with higher values indicating higher mRNA expression. * *p* < 0.05 (*t* test).

**Figure 8 microorganisms-12-00316-f008:**
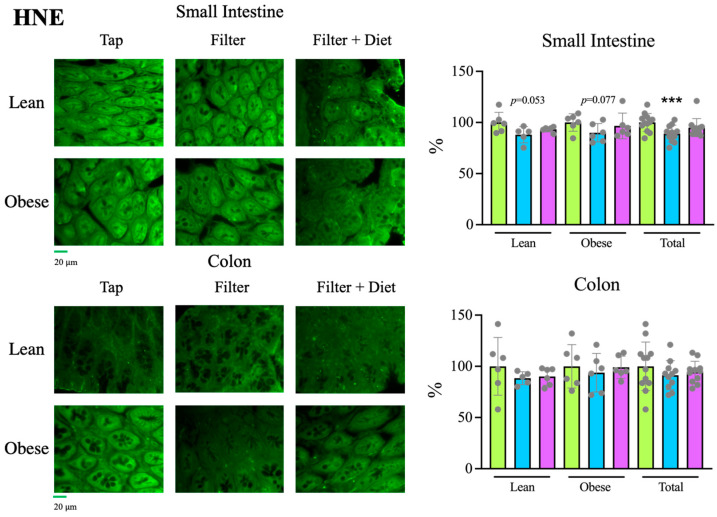
Determination of 4-HNE-modified protein levels by IF analysis of tissue sections from the small intestine and colon. 
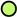
 Tap, 
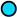
 filter, 
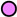
 filter + probiotic. Collected data correspond to the integrated fluorescence signal divided by the tissue area. The graphs show the % relative to the mean of the control group (tap), mean +/− SD. *** *p* < 0.001 (*t* test). Each data point corresponds to the mean of the values obtained for each rat. Representative images captured using a 20× objective are included in the left panels.

**Figure 9 microorganisms-12-00316-f009:**
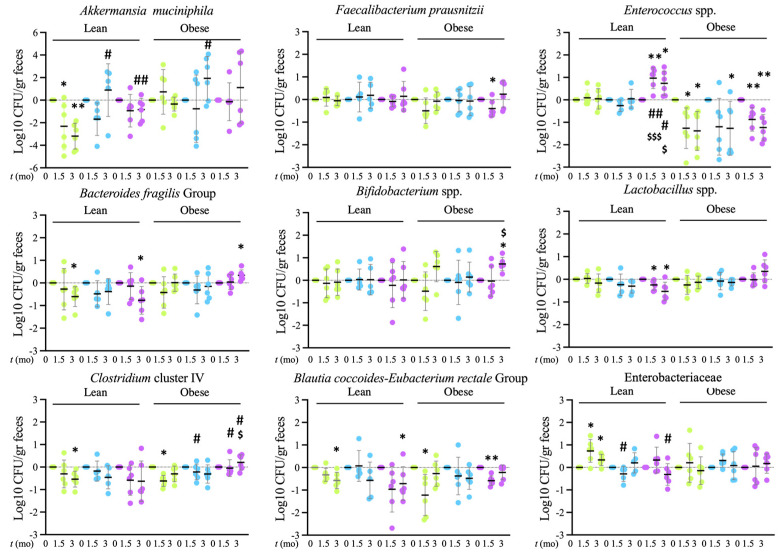
qPCR analysis of microbial content in feces. The graphs show the analysis of bacterial groups analyzed at the indicated times. 
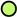
 Tap, 
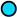
 filter, 
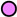
 filter + probiotic. The represented values in the graphs are the differences in the number of bacteria between the values obtained at the indicated time and *t* = 0 for each individual rat analyzed, mean values +/− SD. Statistical comparison between groups was carried out using *t* test type 1 (variations over time) and type 2 (differences between groups). * *p* < 0.05, ** *p* < 0.01 for the comparison with *t* = 0, # *p* < 0.05, ## *p* < 0.01 for the comparison with the control group (tap), $ *p* < 0.05, $$$ *p* < 0.001 for the comparison of the filtered water group with the filter + probiotic group.

## Data Availability

Data supporting reported results are included as Appendix A.

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
