# Peer review of "Intestinal Effects of Filtered Alkalinized Water in Lean and Obese Zucker Rats"

_microorganisms, 2024, doi:10.3390/microorganisms12020316_

Round 1

Reviewer 1 Report

Comments and Suggestions for Authors

Dear María Monsalve

Microorganisms

Manuscript ID: microorganisms-2783854

Special Issue: Functional Foods, Prebiotics and Probiotics Strengthening

Intestinal Microbiome Health

Section: Gut Microbiota

Title: “Intestinal effects of filtered alkalinized water in lean and obese

Zucker rats“.

Major Comments

Materials and Methods

1. “cfu/g” please write in details.  

2. “Fresh feces were 96 collected at t=0, 6 weeks (6w) and three months (3mo) of treatment.” The authors should clarify why these times are chosen for collection.

3. The authors should clarify why did not use the fourth group fed only a diet supplemented with 3 g/kg of Megaflora 9 EVO (Solchem®). i.e. four experimental groups should be assigned in the present study to assess the effects of Megaflora 9 EVO alone.

Results

4. A probiotic (Megaflora 9 EVO) was included in the diet of a group of rats drinking filtered water, adjusting the dose to that recommended for humans. Please provide the human dose with references.

5. In the introduction, the new and original studies should be stated. In addition, the differences from previous studies should be reviewed.

6. The manuscript is interesting. However, authors should inform readers how to scale up the system for practical use.

7. The mechanisms of improvement should be elucidated based on the results.

8. In its current state, the level of English throughout your manuscript does not meet the journal’s required standard. You may wish to ask a native speaker to check your manuscript for grammar, style, and syntax.

Comments on the Quality of English Language

In its current state, the level of English throughout your manuscript does not meet the journal’s required standard. You may wish to ask a native speaker to check your manuscript for grammar, style, and syntax.

Author Response

Reviewer 1. Thank you for helping us improve the manuscript.

Q1. “cfu/g” please write in details.  

Answer. We have modified the text as suggested, it now reads:

“colony forming units (cfu)/g”. (page 3, paragraph 1, line 107).

Q2. “Fresh feces were collected at t=0, 6 weeks (6w) and three months (3mo) of treatment.” The authors should clarify why these times are chosen for collection.

Answer. We have modified the text as suggested, it now includes in the first paragraph of the Results section the following clarification:

“The decision to perform a 3mo treatment (3mo) was based on our previous studies showing significant changes in rats treated for 3 mo. We also collected blood and stool samples at 6 weeks for a better follow up of the treatment responses.” (page 5, paragraph 4, lines 228-231).

Q3. The authors should clarify why did not use the fourth group fed only a diet supplemented with 3 g/kg of Megaflora 9 EVO (Solchem®). i.e. four experimental groups should be assigned in the present study to assess the effects of Megaflora 9 EVO alone.

Answer. There were two main reasons, the funds and the facilities. The available funds did not allow us to include a fourth group. Our animal facilities are currently only housing mice and the space where rats could be allocated was very small. We understand it limits the conclusions of the study and, therefore, we clearly stated that fact in the manuscript.

Q4. A probiotic (Megaflora 9 EVO) was included in the diet of a group of rats drinking filtered water, adjusting the dose to that recommended for humans. Please provide the human dose with references.

Answer. We have modified the text as suggested, including the corresponding reference, it now reads:

“In humans, the recommended dose is 1-2 g [16].” (page 3, paragraph 1, line 110).

Q5. In the introduction, the new and original studies should be stated. In addition, the differences from previous studies should be reviewed.

Answer. We do not understand well what the reviewer is referring to in this case, since in the introduction we already referred to the studies available on both electrolyzed and filtered alkalinized water. Trying to answer this question we have not extended the text with a more detailed description of the available studies using EARW as follows:

“A two week intervention study with EARW in healthy human volunteers concluded that water pH had no impact on the composition of the gut microbiota or glucose regulation in young male adults. [11]. In contrast, in another study, mice were treated for four weeks with EARW and the authors showed that in the fecal microbiome the relative abundances of 20 taxa differed significantly from controls. However, the significance of these changes was not stablished [12].” (page 2, paragraph 2, lines 56-62).

Q6. Authors should inform readers how to scale up the system for practical use.

Answer. We have modified the closing conclusion paragraphs aiming to answer this comment. The text now reads:

“The diet supplementation with probiotic had a distinct impact on the microbiome that was clearly independent from the effect of the water, although further testing of a group treated only with the probiotic would be necessary to draw definite conclusions (Figure 9). Nevertheless, since no clear indication of a cooperative or cancelling cross-effect was identified, it possibly suggests that alkalinized filtered water and the tested probiotics can be used safely in combination.” (page 14, paragraph 5, lines 528-533).

“Therefore, these data support the hypothesis that drinking filtered alkalinized water can improve the intestinal inflammatory state and reduce dysbiosis, resulting in a better state of general health, although human studies will still be necessary to confirm the significance of these findings.” (page 14, paragraph 6, lines 534-537).

Q7. The mechanisms of improvement should be elucidated based on the results.

Answer. We think at this stage the results do not allow us to make strong assertions regarding the mechanisms involved, nevertheless we have now included what we think is the most plausible mechanism of action in the discussion section including the following text:

“Although highly tentative at this stage, the derived data suggest that alkalinized filtered water is possibly less irritating to the intestinal mucosa than tap water and it may also have a similar direct effect on the intestinal bacteria, helping to preserve intestinal homeostasis, and hence prevent systemic inflammation and boost metabolic homeostasis.” (page 14, paragraph 4, lines 524-527).

Q8. In its current state, the level of English throughout your manuscript does not meet the journal’s required standard. You may wish to ask a native speaker to check your manuscript for grammar, style, and syntax.

Answer. We will make use of the journal’s editing service, the time allocated to us for this resubmission did not allow as sufficient time span to submit the text to an external English editing service. At this stage we have just performed an in-house revision looking for typos and grammar errors.

Reviewer 2 Report

Comments and Suggestions for Authors

The manuscript titled "Intestinal Effects of Filtered Alkalinized Water in Lean and Obese Zucker Rats" presents original and significant research with strong scientific foundations. The originality is high, as it explores a unique aspect of the effects of alkalinized water on gut health, an area not extensively covered in existing literature. The content is significant, offering potentially impactful insights into gastrointestinal health and obesity-related issues. The presentation quality and scientific soundness is average, suggesting room for improvement in terms of clarity and accessibility to a broader audience. Interest to readers is average, as the topic, while highly relevant to specialists, might not engage a wider audience. Overall, the manuscript's merit is average, contributing relatively valuable knowledge to the field and demonstrating a well-executed study.

The "Methods" section of your manuscript can be improved by enhancing the clarity and detail of the descriptions provided. Specifically, additional elaboration on the experimental procedures, including more precise details about the methods used for assessing the effects of alkalinized water, would be beneficial. This could include more information on the control conditions, the specifics of the water filtration and alkalinization processes, and the statistical methods used for data analysis. Providing these details would enhance the replicability of the study and strengthen the scientific rigor of the paper. The methods section lacks sufficient detail, which raises questions about the reproducibility of the study. A more comprehensive description of the experimental procedures, including specifics about the water treatment process and statistical analysis, is necessary for a study of this nature.

Few areas that could be improved:

Typos and Grammatical Errors: Some minor typographical and grammatical errors were noticed. There are instances of missing commas and minor punctuation inconsistencies.  I recommend a detailed, line-by-line reading of the manuscript, preferably by a professional editor or a peer with expertise in the field. You may utilize MDPI service for that

Acronym Clarification: While most acronyms are well-defined, ensuring that each acronym is defined upon its first use in each major section can improve clarity, especially for readers who might skip to specific sections of interest.

Consistency in Terminology: Ensure consistent use of terms throughout the manuscript. For example, if you're referring to "alkalinized water" or "alkaline water," stick to one term throughout to avoid confusion.

Formatting Inconsistencies: Some formatting irregularities, such as inconsistent spacing and alignment in the text and figures, can be polished for a more professional presentation.

Also, this research, while scientifically sound, may not attract a wide readership. Its specific focus on alkalinized water and its effects on Zucker rats might not be of significant interest to a broader audience.

Based on the review of the manuscript "Intestinal Effects of Filtered Alkalinized Water in Lean and Obese Zucker Rats," my decision would be:

Accept after minor revision (corrections to minor methodological errors and text editing)

This decision is based on the good quality of the originality, significance, and scientific soundness of the work, combined with the need for minor improvements in clarity and detail in the methods section, as well as some textual and formatting refinements. These revisions should enhance the overall quality of the manuscript without necessitating a complete overhaul or additional experiments.

Happy upcoming holidays!

Author Response

Reviewer 2. Thank you for helping us improve the manuscript.

Q1. The "Methods" section can be improved by enhancing the clarity and detail of the descriptions provided. Specifically, additional elaboration on the experimental procedures, including more precise details about the methods used for assessing the effects of alkalinized water, would be beneficial. This could include more information on the control conditions, the specifics of the water filtration and alkalinization processes, and the statistical methods used for data analysis.

Answer. We have now included additional methodology details as follows:

“Indirect calorimetry…. Data presented correspond to the mean values​​ for O2 consumed or CO2 produced per hour and registered during the day (12 h) or night (12 h) periods of the second 24-h cycle (2nd day), when the measurements were more stable.” (page 3, paragraph 2, lines 129-132).

Regarding the filtering system used we have. Now included the following text with references:

“The water filtering system has been analyzed by an authorized laboratory (Oliver Rodés) and found to conform with UNE 149101:2015 norm for human consumption the filtering system has also been shown to remove Trihalomethanes (THMs) and Cl- while adding 15 mg/L of Mg2+, it also has been shown to provide water free of microorganisms, and not to release Na+. The plastic jar has been approved for contact with food and demonstrated to be free of Bisphenol A (BPA), Epoxidized soybean oil (ESBO) and phthalate esters (16). (from page 1, paragraph 6, line 99 to page 2, paragraph 1, line 105).

Regarding the statistical analysis, for further clarity the statistical analysis paragraph has been extended as follows:

“Statistical significance of differences between groups was assessed using two-tailed unpaired or paired t-test for comparisons between two groups, one-way and two-way ANOVA was used for the analysis of time dependent variation and the differential response to treatments of two groups respectively.” (page 5, paragraph 4, lines 216-220).

Additional details have also been included in the figure legends to clarify on how data was analyzed as follows:

Figure 1. Absolute (upper panel) and relative (lower panel) weight gain determined every two weeks for 3mo. In the bottom panel, weight values at t = 0 were considered 100%.” (page 6, paragraph 1, lines 242-243).

“Figure 2. Changes in plasma levels of IL-1b and IL-10 (levels at 6w – levels at t = 0) following 6w of treatment.” (page 6, paragraph 3, lines 256-257).

Figure 3. Electrochemical analysis of the total antioxidant capacity (QT) in plasma samples from rats treated for 6w (left panel) or 3mo (right panel).   Tap,   Filter,   Filter + Probiotic. The graph shows the electrochemical units provided by the e-BQC reader and includes mean values ​​+/- SD.” (page 7, paragraph 2, lines 266-268).

Figure 4.…. The values shown in the graphs are the difference of O2 volume (V) consumed (upper panel) or CO2 produced (lower panel) between the indicated time and t= 0 for each individual rat analyzed. Determinations were taken every hour for 72 h, the presented data corresponds to the mean of both 12 h day and 12 h night periods of the 2nd day, when collected values were more stable.” (from page 7, paragraph 4, line 288 to page 8, paragraph 1, line 292).

“Figure 5. Determination of MUC2 levels by IF analysis of tissue sections from the small intestine and colon.   Tap,   Filter,   Filter + Probiotic. Collected data correspond to the integrated fluorescence signal divided by the tissue area. The graphs show the x fold change ​​relative to the mean of the lean or obese control group (tap), mean ​​+/- SD. *p<0.05 (t test).  Each data point corresponds to the mean of the values obtained for each rat…”. (page 8, paragraph 3, lines 311-314).

“Figure 6. … Collected data correspond to the positive area divided by the tissue area The graphs show the % ​​relative to the mean of the control group (tap), mean ​​+/- SD. *p<0.05 (t-test). Each data point corresponds to the mean of the values obtained for each rat….”. (page 9, paragraph 3, lines 332-335).

“Figure 7. …The graphs show the differences in DDCt of the individual values per rat and the mean value of the corresponding control group (tap), mean +/- SD. For clarity sake, -DDCt values are presented, with higher values indicating higher mRNA expression…”. (page 10, paragraph 1, lines 347-350).

“Figure 8. …Collected data correspond to the integrated fluorescence signal divided by the tissue area. The graphs show the % ​​relative to the mean of the control group (tap), mean ​​+/- SD. *p<0.05 (t test). Each data point corresponds to the mean of the values obtained for each rat. Representative images captured using a 20x objective are included in the left panels. mean ​​+/- SD. *p<0.05 (t-test)….” (from page 10, paragraph 3, line 364 to page 11, paragraph 1, line 368).

“Figure 9… The represented values in the graphs are the differences in the number of bacteria between the values obtained at the indicated time and t= 0 for each individual rat analyzed...”. (page 12, paragraph 3, lines 434-436).

Q2. Typos and Grammatical Errors: Some minor typographical and grammatical errors were noticed. There are instances of missing commas and minor punctuation inconsistencies.  I recommend a detailed, line-by-line reading of the manuscript, preferably by a professional editor or a peer with expertise in the field. You may utilize MDPI service for that.

Answer. We will make use of the journal’s editing service, the time allocated to us for this resubmission did not allow as sufficient time span to submit the text to an external English editing service. At this stage we have just performed an in-house revision looking for typos and grammar errors.

Acronym Clarification: While most acronyms are well-defined, ensuring that each acronym is defined upon its first use in each major section can improve clarity, especially for readers who might skip to specific sections of interest.

Answer. We have revised the text and hopefully fixed this issue.

Consistency in Terminology: Ensure consistent use of terms throughout the manuscript. For example, if you're referring to "alkalinized water" or "alkaline water," stick to one term throughout to avoid confusion.

Answer. We have revised the text and hopefully fixed this issue.

Formatting Inconsistencies: Some formatting irregularities, such as inconsistent spacing and alignment in the text and figures, can be polished for a more professional presentation.

Answer. We have revised the text and hopefully fixed this issue.

Round 2

Reviewer 1 Report

Comments and Suggestions for Authors

Dear María Monsalve

Microorganisms

Manuscript ID: microorganisms-2783854 v2

Special Issue: Functional Foods, Prebiotics and Probiotics Strengthening

Intestinal Microbiome Health

Section: Gut Microbiota

Title: “Intestinal effects of filtered alkalinized water in lean and obese

Zucker rats“.

Congratulation

Well done.

Thank you very much.